# Evaluating CRISPR-based prime editing for cancer modeling and CFTR repair in organoids

Maarten H Geurts[1,2], Eyleen de Poel[3,4,*], Cayetano Pleguezuelos-Manzano[1,2,*], Rurika Oka[5], Léo Carrillo[1,2], Amanda Andersson-Rolf[1,2], Matteo Boretto[1,2], Jesse E Brunsveld[3,4], Ruben van Boxtel[5], Jeffrey M Beekman[3,4], Hans Clevers[1,2]

**Prime editing is a recently reported genome editing tool using a nickase-cas9 fused to a reverse transcriptase that directly synthesizes the desired edit at the target site. Here, we explore the use of prime editing in human organoids. Common TP53 mutations can be correctly modeled in human adult stem cell–derived colonic organoids with efficiencies up to 25% and up to 97% in hepatocyte organoids. Next, we functionally repaired the cystic fibrosis CFTR-F508del mutation and compared prime editing to CRISPR/Cas9–mediated homology-directed repair and adenine base editing on the CFTR-R785* mutation. Whole-genome sequencing of prime editing–repaired organoids revealed no detectable off-target effects. Despite encountering varying editing efficiencies and undesired mutations at the target site, these results underline the broad applicability of prime editing for modeling oncogenic mutations and showcase the potential clinical application of this technique, pending further optimization.**

## Introduction

The field of genome engineering has been revolutionized by the development of the efficient genome editing tool CRISPR/Cas9. In CRISPR/Cas9–mediated genome engineering, the effector protein cas9 is guided towards the target site in the genome by an RNA guide (Jinek et al, 2012). Upon target recognition, cas9 generates a double stranded break (DSB) that can be exploited for a variety of genome engineering strategies (Cong et al, 2013; Mali et al, 2013). Because of the easy reprogrammability and high efficiency of CRISPR/Cas9, the technology is widely used for gene modification and is considered to be the most promising tool for clinical gene editing. However, the repair of DSBs is often error-prone and can result in unwanted DNA damage at the target site as well as at

off-target sites that closely resemble the guide-RNA (Fu et al, 2013; Pattanayak et al, 2013; Cho et al, 2014; Kosicki et al, 2018). These issues have been circumvented by the development of Cas9 fusion proteins, called base editors. In base editing, a partially nuclease-inactive nickase-cas9 (nCas9) protein is fused to either the cytidine deaminase APOBEC1A to enable C-G to T-A base pair changes or to an evolved TadA heterodimer to facilitate the opposite reaction, turning A-T base pairs into G-C base pairs (Komor et al, 2016; Gaudelli et al, 2017). Base editors show high efficiency and infrequent unwanted DNA changes in a variety of model systems but are strictly limited to transition DNA substitutions (Pavlov et al, 2019; Zuo et al, 2019; Geurts et al, 2020).

To overcome these limitations, prime editing has been developed to enable both transition and transversion reactions as well as insertions and deletions of up to 80 nucleotides in length without the need to generate DSBs (Anzalone et al, 2019). In prime editing, an nCas9 is fused to an engineered reverse transcriptase (RT) that is used to generate complementary DNA from an RNA template (PE2) (Fig 1). This fusion protein is combined with a prime editing guide-RNA (pegRNA) that guides the nCas9 to its target and contains the RNA template that encodes the desired edit. Upon target recognition the protospacer adjacent motif (PAM)–containing strand is nicked and the pegRNA extension binds to the nicked strand at the primer-binding site (PBS). The RT domain then uses the remainder (RT template) of the pegRNA to synthesize a 3′-DNA-flap containing the edit of interest. This DNA-flap is resolved by cellular DNA repair processes that can be further enhanced by inducing a proximal second nick in the opposing DNA strand, guided by a second (PE3) guide-RNA (Anzalone et al, 2019) (Fig 1). Prime editing holds great promise, as it can—in theory—repair 89% of all disease-causing variants (Anzalone et al, 2019). Here, we apply this approach in human organoids to introduce cancer mutations and to repair mutations in the cystic fibrosis transmembrane conductance regulator (CFTR) channel that cause cystic fibrosis (CF), a Mendelian disorder with high prevalence in European ancestry.

[1]Hubrecht Institute, Royal Netherlands Academy of Arts and Sciences (KNAW) and University Medical Center Utrecht, Utrecht, the Netherlands  [2]Oncode Institute, Hubrecht Institute, Utrecht, the Netherlands  [3]Department of Pediatric Respiratory Medicine, Wilhelmina Children's Hospital, University Medical Center, Utrecht University, Utrecht, the Netherlands  [4]Regenerative Medicine Utrecht, University Medical Center, Utrecht University, Utrecht, the Netherlands  [5]Oncode Institute,Princes Maxima Center, Utrecht, The Netherlands

Correspondence: h.clevers@hubrecht.eu
*Eyleen de Poel and Cayetano Pleguezuelos-Manzano contributed equally to this work

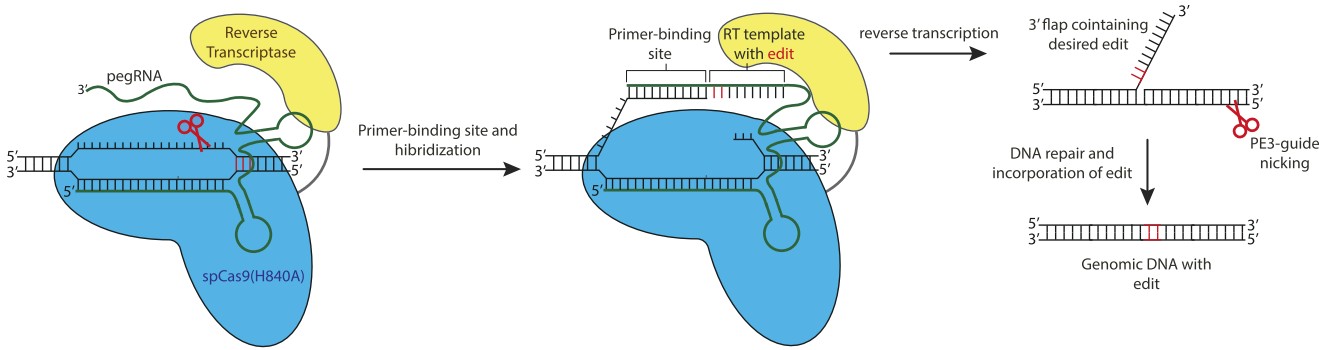

**Figure 1. Principles of prime editing adapted from Anzalone et al (2019).**
Principles of prime editing: The pegRNA complexes with the nCas9 (H840A)–reverse transcriptase (RT) prime-editing fusion protein and binds to the target DNA. Upon protospacer adjacent motif strand cleavage by nCas9, the primer-binding site of the pegRNA extension binds the single-stranded DNA upon which the RT synthesizes a 3′-DNA flap containing the edit of interest. This 3′-flap is resolved by cellular DNA processes which can be further enhanced by introducing a proximal second nick in the opposing DNA strand, guided by a second (PE3) guide-RNA. Red scissors indicate nick site of the nCas9. RT = Reverse Transcriptase.

# Results

## Modeling common mutations in cancer in colon and hepatocyte organoids

We first characterized and optimized prime-editing efficacy in adult human stem cell–derived organoids by targeting *TP53*, a gene that is often mutated in cancer. Previously, we have shown that *TP53*-mutant organoids can be selected by adding nutlin-3, a molecule that inhibits the interaction between TP53 and MDM2, to the organoid culture medium (Drost et al, 2015; Matano et al, 2015). By co-transfecting plasmids containing genome-editing components targeting *TP53* with plasmids encoding a PiggyBac system conveying hygromycin resistance to transfected organoids, we can simultaneously functionally detect TP53 mutants by nutlin-3 resistance and determine editing efficiency by Sanger sequencing of hygromycin-resistant clones (Fig 2A). Using the pegFinder online software tool, we designed a single pegRNA and PE3-guide pair to introduce the R175H mutation, the most common mutation found in TP53 according to the Catalogue Of Somatic Mutations in Cancer (Forbes et al, 2017; Chow & Chen, 2020 *Preprint*). The pegRNAs were designed to integrate a PAM-disrupting mutation to block re-binding of Cas9 after the correct editing event has occurred. We co-transfected PE2 plasmids, the pegRNA/PE3-guide pair and hygromycin resistance PiggyBac plasmids in colonic organoids by electroporation. Clonally selected organoids appeared after 2 wk of nutlin-3 selection whereas control organoids, transfected with PE2 plasmids and a non-targeting scrambled sgRNA did not grow out (Fig 2B). Manual picking of selected organoids and subsequent sanger sequencing showed correct homozygous induction of the *TP53*-R175H (c.524 G>A) mutation in seven of the eight clonally expanded colonic organoids (Fig 2C).

To determine editing efficiency, we performed Sanger sequencing on 36 hygromycin-selected colonic clones and found a single organoid harboring a heterozygous R175H mutation (Fig 2D). Next, we performed the same experiment in hepatocytes and found a significantly increased editing efficiency clearly shown from nutlin-3 selection (Fig 2B). Of 36 hygromycin-selected clones (Fig 2D), 20 (55.5%) harbored a homozygous mutation, 4 (11.2%) a heterozygous mutation and two remained WT (5.5%) (Fig 2D and E). The

remaining 10 clones (27.8%) had incorporated unintended DNA changes around the target side, caused by incorrect repair of either pegRNA or PE3-guide nicking (Fig S1A).

The most common mutation in *TP53* in hepatocellular carcinoma is R249S, most prominently caused by Aflatoxin B1. Exposure by this carcinogen that is often found in contaminated food sources results in the c.747 G>T transversion (Aguilar et al, 1993). To generate hepatocyte and colonic organoids harboring this mutation, we used the pegFinder online software tool to design a single pegRNA and PE3-guide pair to introduce the *TP53*-R249S mutation. No additional PAM disrupting mutation was designed as the G>T transversion itself would disrupt the PAM. We co-transfected PE2 plasmids, the pegRNA/PE3-guide pair, and hygromycin resistance PiggyBac plasmids in hepatocyte and colon organoids by electroporation. Similar to the previously described experiment, clonal survival and outgrowth after 2 wk of nutlin-3 selection showed an efficient mutation induction in hepatocyte organoids, whereas the induction of mutations was less efficient in colon organoids (Fig 2D). Of 36 hygromycin-selected hepatocyte organoids, prime editing induced homozygous R249S mutations in 35 clones (97.2%), whereas a heterozygous mutation was observed in 1 clone (2.8%) (Fig 2E and F). Editing efficiencies in colon organoids were significantly lower, 8 organoids contained heterozygous mutations (22.2%), whereas the other 28 remained wild type (77.8%) (Fig 2E). Thus, prime editing induced mutations in 25% of the intestinal organoids, yet prime editing was far more efficient in hepatocytes on this target.

Next, we aimed to construct five additional mutations that are commonly found in *TP53* (Fig S2A). Only the pegRNA/PE3-guide pair designed for the induction of *TP53*-C176F resulted in clones capable of surviving nutlin-3 selection whereas the other pairs did not (Fig S2B). Manual picking of these clones followed by Sanger sequencing showed correct homozygous introduction of the C176F (c.527 G>T) mutation in the *TP53* gene including the designed PAM disruption mutation (Fig S2C). Sanger sequencing of 36 hygromycin-resistant clones revealed only a single clone that contained a heterozygous C176F mutation, indicating a low editing efficiency at this target site (Figure S2D). In addition, we designed pegRNA/PE3-guide pairs to generate mutations in *APC*, the gene that is often the first to be mutated in colorectal cancer (Fig S2A). Mutations in *APC*

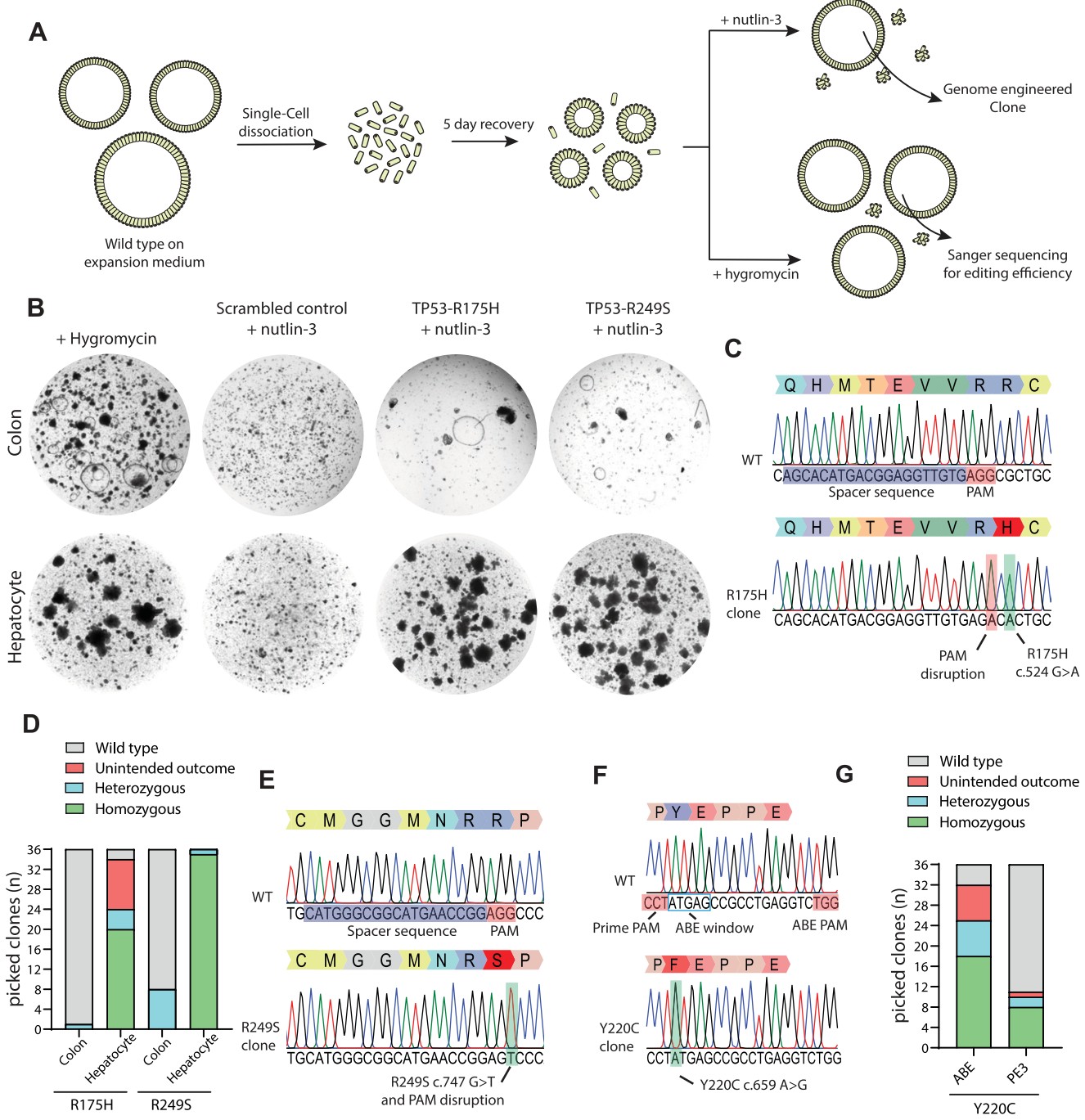

**Figure 2.  Prime editing enables generation of oncogenic mutations in organoids.**
**(A)** Strategy to generate *TP53*-mutated human organoids. **(B)** Bright-field images of prime-editing experiments targeting the *TP53*-R175H and *TP53*-R249S mutations compared with a negative scrambled sgRNA control and hygromycin resistance. **(C)** Sanger sequencing trace of selected clonal organoids harboring the *TP53*-R175H mutation compared with WT. **(D)** Prime-editing efficiency on *TP53*-R175H and *TP53*-R249S as determined by Sanger sequencing on hygromycin-resistant clones. **(E)** Sanger sequencing trace of selected clonal organoids harboring the *TP53*-R249S mutation compared with WT. **(F)** Sanger sequencing trace of selected clonal organoids harboring the *TP53*-Y220C mutation compared with WT. **(G)** Adenine base editing versus prime-editing efficiency on the *TP53*-Y220C mutation as determined by Sanger sequencing of hygromycin-selected clones. Protospacer adjacent motifs are shown in red and guide-RNA sequences are shown in blue.

can be selected in culture by the removal of the expansion medium components WNT and R-spondin (Drost et al, 2015; Matano et al, 2015). Selection for *APC* mutants by removal of WNT and R-spondin after transfection of two pegRNA/PE3-guide pairs resulted in

outgrowth of a single clone (Fig S2E). Interestingly, instead of the designed *APC* R1450* (c.4348C > T) mutation, Sanger sequencing revealed a homozygous duplication of the 37 nucleotides directly upstream of the single stranded nick introduced by the SpCas9

(H840A) (Fig S2F). These results indicated that prime editing can induce mutations in intestinal and hepatocellular adult human stem cells at varying efficiencies but may yield undesired outcomes.

## Prime editing versus adenine base editing

To directly compare prime editing to base editing we focused on the *TP53*-Y220C (c.659A > G) mutation. As this is an A>G transition reaction, it can be modeled by both adenine base editing and prime editing. A sgRNA for adenine base editing could be designed with the A on position 4 of the sgRNA and position 1 in the editing window, whereas a PAM on the opposite site of the intended mutation could be exploited for prime editing (Fig 2F). We co-transfected either prime editing- or base editing constructs with the hygromycin resistance PiggyBac system into colon organoids. To compare editing efficiency of adenine base editing versus prime editing, we Sanger-sequenced 36 hygromycin resistant colon organoid clones from both transfections. Adenine base editing resulted in correct homozygous Y220C induction in 50% of the clones and an additional seven clones (19.6%) that harbored a correct heterozygous mutation (Fig 2G). A further seven clones (19.6%) had either undergone correct homozygous or heterozygous mutation induction but also harbored an additional A>G transition of the A on position 7 sgRNA (position 4 of the editing window), resulting in the unintended E221G mutation on top of Y220C (Fig S1B). Prime editing was less efficient on this target as we observed eight clones (22.2%) with homozygous and two clones (5.6%) with heterozygous mutation induction. Of 36 clones, only 1 clone (2.8%) harbored an unintended editing outcome underscoring previous observations that, compared with prime editing, base editing is more efficient but the application can be limited by additional editable residues within the editing window.

## Repair of *CFTR*-F508del mutation using prime editing in intestinal organoids

Intestinal organoids are a suitable in vitro disease model of CF as fluid transport into the organoid lumen is fully dependent on the activity of the CFTR channel, stimulated by a rise in forskolin-induced intracellular cAMP levels. Wild-type organoids show a forskolin-induced swelling (FIS) response, whereas organoids derived from people with CF, expressing less functional CFTR protein, show a strongly reduced FIS response (Dekkers et al, 2013). This in vitro assay enables the prediction of in vivo drug response and is clinically applied to tailor treatment for individuals with CF in The Netherlands (Berkers et al, 2019). Previously, we have shown that we can use this FIS response as a direct functional readout for repair of the *CFTR* gene in organoids derived from CF patients, both by base editing and by classical CRISPR-mediated homology-dependent repair (HDR) (Schwank et al, 2013; Geurts et al, 2020). The most common *CFTR* mutation F508del cannot be repaired by base editors. Although CRISPR/Cas9–mediated HDR has been used to repair this mutation in organoids, editing efficiency was low (Schwank et al, 2013). As such, we pursued prime editing–mediated repair of the *CFTR*-F508del mutation, by transfecting intestinal CF organoids carrying the homozygous *CFTR*-F508del mutation with pegRNA/PE3-guide pairs (Fig 3). Forskolin treatment 2 wk after electroporation

showed a swelling response in a single transfected organoid (Fig 3A and B). PCR amplification of the target site, followed by subcloning and Sanger sequencing revealed heterozygous repair of the *CFTR*-F508del mutation in this selected clone (Fig 3C). We tried to further optimize prime editing by designing additional pegRNA/PE3-guide pairs with varying RT and PBS lengths and distance between the pegRNA and PE3-guide as these variables greatly impact editing efficiencies (Anzalone et al, 2019) (Fig S3A). We compared editing efficiencies of eight different combinations of pegRNA/PE3-guide pairs (PBS length = 14 or 15 nucleotides, RT length = 17 or 37 nucleotides, distance to PE3 nick, 63, 21, 41, and 82) directly to conventional CRISPR/Cas9–mediated HDR by counting forskolin-responsive organoids derived from two donors (Fig S3B). *CFTR*-F508del repair by CRISPR/Cas9–mediated HDR resulted in 108 and 124 FIS responsive clones depending on the donor, whereas prime editing never resulted in more than four repaired organoids, indicating low prime-editing efficiencies at this target site (Figs 3D and S3B). No significant differences on the number of repaired clones were observed between the two donors. Repaired clonal organoid lines generated by prime editing and CRISPR/Cas9–mediated HDR exhibited FIS at WT levels or higher, indicating complete functional repair of CFTR function in these organoids. As expected, unrepaired clones did not respond to forskolin (Fig 3E–G). Sanger sequencing followed by deconvolution of the Sanger traces of two additional prime-edited clones and one clone repaired by CRISPR/Cas9–mediated HDR showed correct heterozygous repair of the mutation in one prime editing clone. However, the second clone as well as the clone repaired by HDR contained a small indel at the repair site in the second allele (Fig S3C). These results indicated that even though efficiencies are low and undesired outcomes may occur, prime editing can repair the *CFTR*-F508del mutation in patient-derived intestinal organoids.

## Comparison of *CFTR*-R785* repair by prime editing versus repair by base editing

To directly compare prime editing to base editing, we focused on the repair of the *CFTR*-R785* mutation. Previously, we have shown that this mutation is reparable in patient-derived intestinal organoids with an editing efficiency of ~9%, whereas HDR efficiency was below 2% (Geurts et al, 2020). We designed eight pairs of pegRNA/PE3-guides with varying PBS (13 or 18), RT l (27 or 30) lengths and different distances to the PE3 nick (64, 21, 43, and 82) to find optimal prime-editing conditions (Fig S4A). To assess prime-editing efficiencies, we transfected *CFTR*-R785* organoids with a PE2-P2A-GFP plasmid together with our eight pegRNA/PE3-guide pairs in duplicates and selected transfected cells by FACS sorting (Fig S4B). 2 wk after FACS sorting, forskolin-responsive clones were observed and counted (Fig 4A). Most prime-editing conditions resulted in FIS responsive clones, although editing efficiencies differed greatly (between 0 and 5.7% repaired clones) (Figs 4B and S4B). We then compared these editing efficiencies with base editing and CRISPR/Cas9–mediated HDR, using previously established reagents (Geurts et al, 2020). Forskolin treatment revealed an editing efficiency of 9.1% corrected organoids by ABE and 1.22% by CRISPR/Cas9–mediated HDR (Figs 4A and S4B). Repaired clonally expanded organoid lines generated by prime editing and base

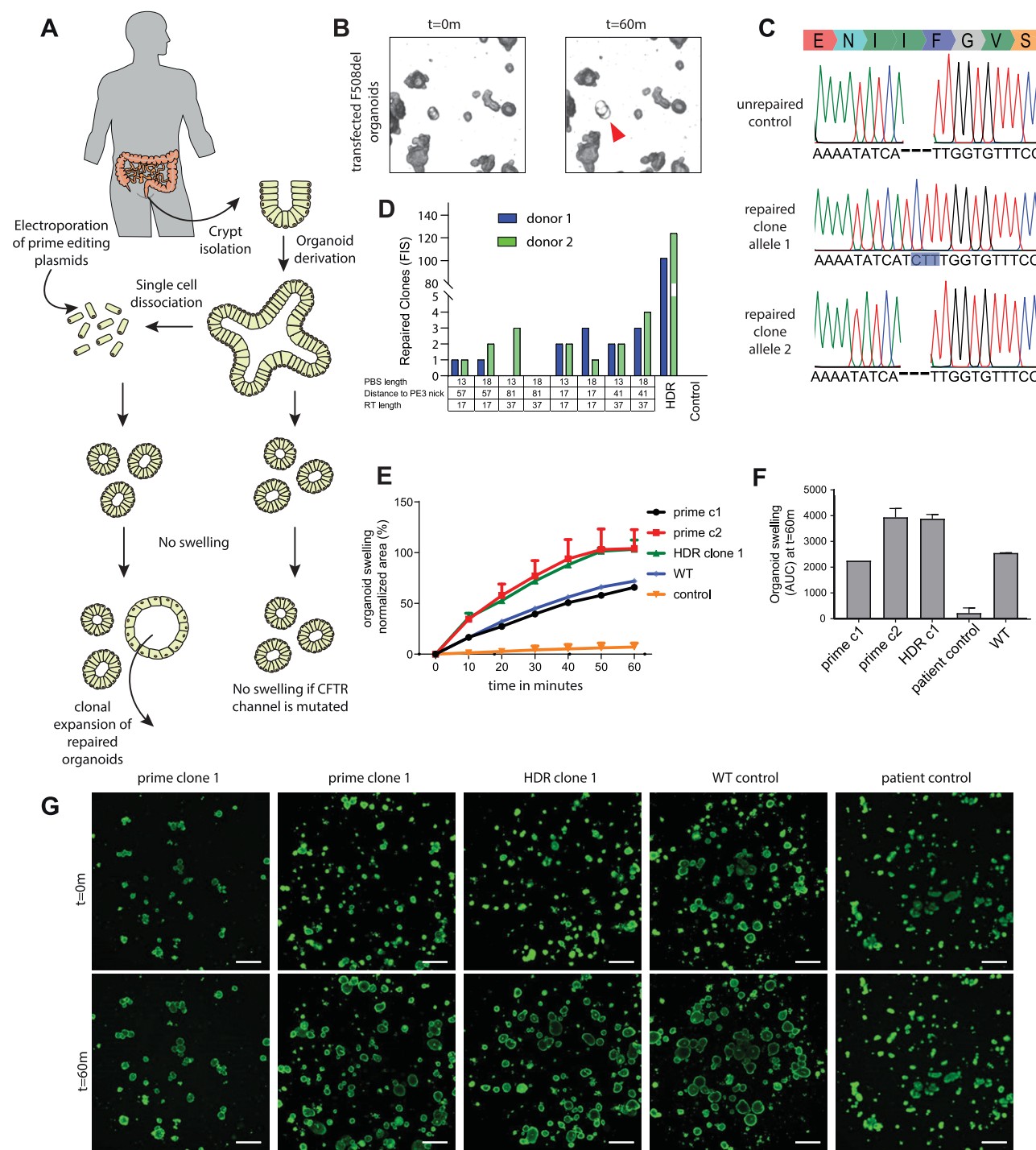

**Figure 3. Functional repair of the *CFTR*-F508del mutation in patient-derived intestinal organoids.**
**(A)** Experimental design of prime editing-mediated repair of *CFTR* mutations in human intestinal organoids. **(B)** Transfected *CFTR*-F508del organoids before (t = 0) and after (t = 60 m) addition of forskolin. Functionally repaired organoid indicated with red arrow. **(C)** Sanger sequencing traces of both alleles of a functionally selected *CFTR*-F508del organoid line compared with unrepaired control organoids. Blue box shows the prime editing–induced insertion. **(D)** Prime-editing efficiencies for the repair of the *CFTR*-F508del mutation in two donors as measured by Forskolin-induced swelling reactive organoids compared with CRISPR/Cas9–mediated homology-dependent repair and a negative scrambled sgRNA control. **(E)** Per well the total organoid area (xy plane in $\mu m^2$) increase relative to t = 0 (set to 100%) of forskolin treatment was quantified (n = 3). **(F)** Forskolin-induced swelling as the absolute area under the curve (t = 60 min; baseline, 100%), mean ± SD; n = 3, *P < 0.001, compared with the corrected organoid clones and the WT organoid sample. **(G)** Confocal images of calcein green–stained patient-derived intestinal organoids before and after 60 min stimulation with forskolin (scale bars, 200 $\mu m$).

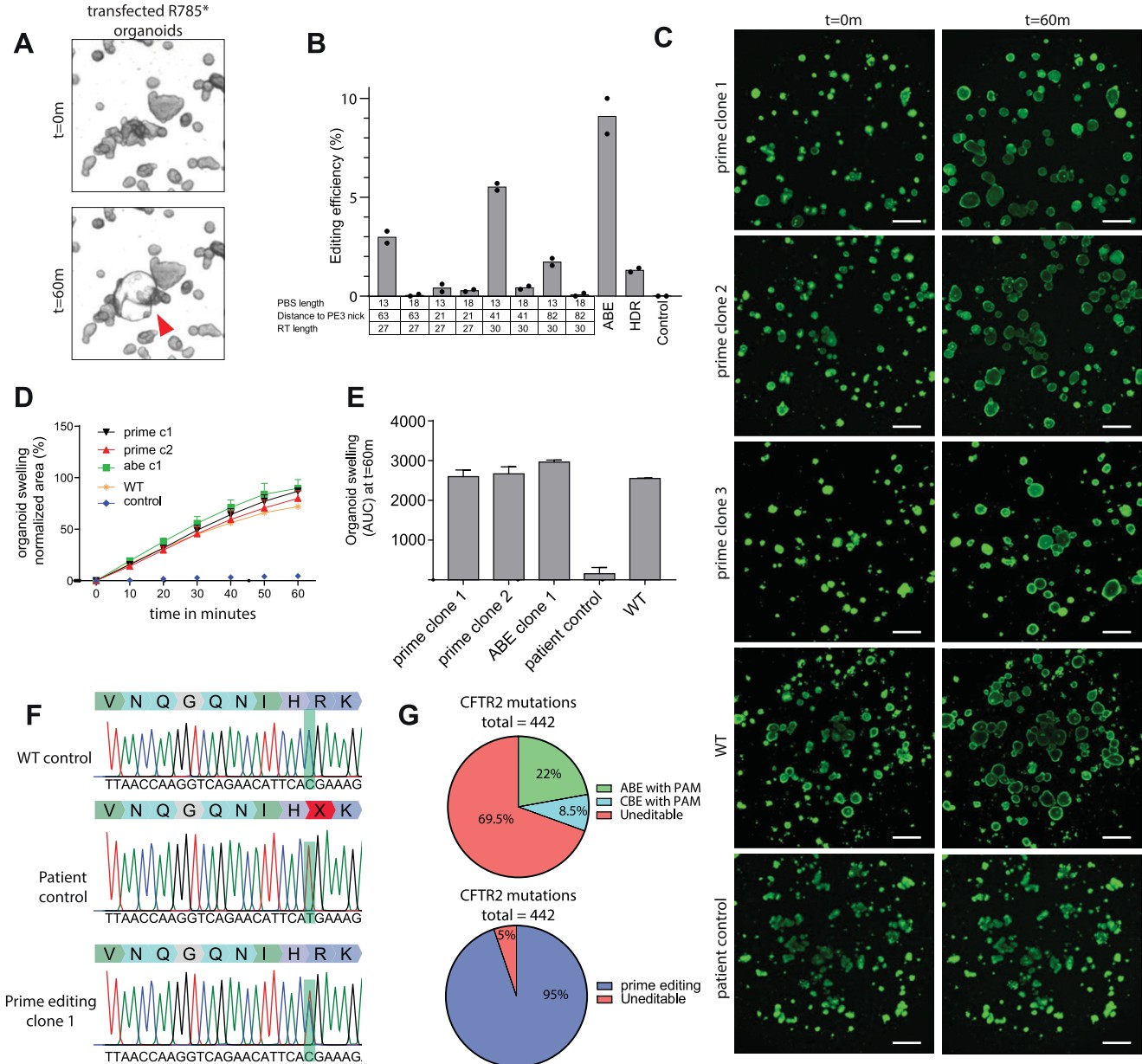

**Figure 4. Functional repair of the *CFTR*-R785\* mutation in patient-derived intestinal organoids.**
**(A)** Transfected *CFTR*-R785\* organoids before (t = 0) and after (t = 60 m) addition of forskolin. Functionally repaired organoid indicated with red arrow. **(B)** Prime-editing efficiencies for the repair of the *CFTR*-R785\* mutation as measured by Forskolin-induced swelling reactive organoids compared with adenine base editing, CRISPR/Cas9–mediated homology-dependent repair and a negative scrambled sgRNA control. **(C)** Confocal images of calcein green–stained patient-derived intestinal organoids before and after 60-min stimulation with forskolin (scale bars, 200 *μ*m). **(D)** Per well the total organoid area (xy plane in *μ*m$^2$) increase relative to t = 0 (set to 100%) of forskolin treatment was quantified (n = 3). **(E)** Forskolin-induced swelling as the absolute area under the curve (t = 60 min; baseline, 100%), mean ± SD; n = 3, *P < 0.001, compared with the corrected organoid clones and the WT organoid sample. **(F)** Sanger sequencing traces of both alleles of a functionally selected *CFTR*-F508del organoid line compared with unrepaired control organoids. Blue box shows the prime editing induced insertion. **(G)** Pie chart showing mutations in *CFTR* that can be targeted by cytosine and adenine base editing compared with prime editing.

editing exhibited a forskolin response similar to WT levels, indicating complete repair of CFTR function in these organoids (Fig 4C–E). Sanger sequencing of three repaired organoid lines showed that two of three clones repaired by prime editing and the ABE clone underwent correct repair of the *CFTR*-R785\* mutation on a single allele, whereas the second allele remained undamaged (Figs 4F and S4C). The third prime-edited clone and the HDR-repaired

clone contained a small indel at the repair site on the second allele, indicating DNA damage (Fig S4C).

These results again underscore that the current version of adenine base editing is superior to prime editing in both safety and efficiency if the mutation is targetable by adenine base editing and no additional editable residues reside within the editing window (Geurts et al, 2020). However, if a mutation is not reparable by base

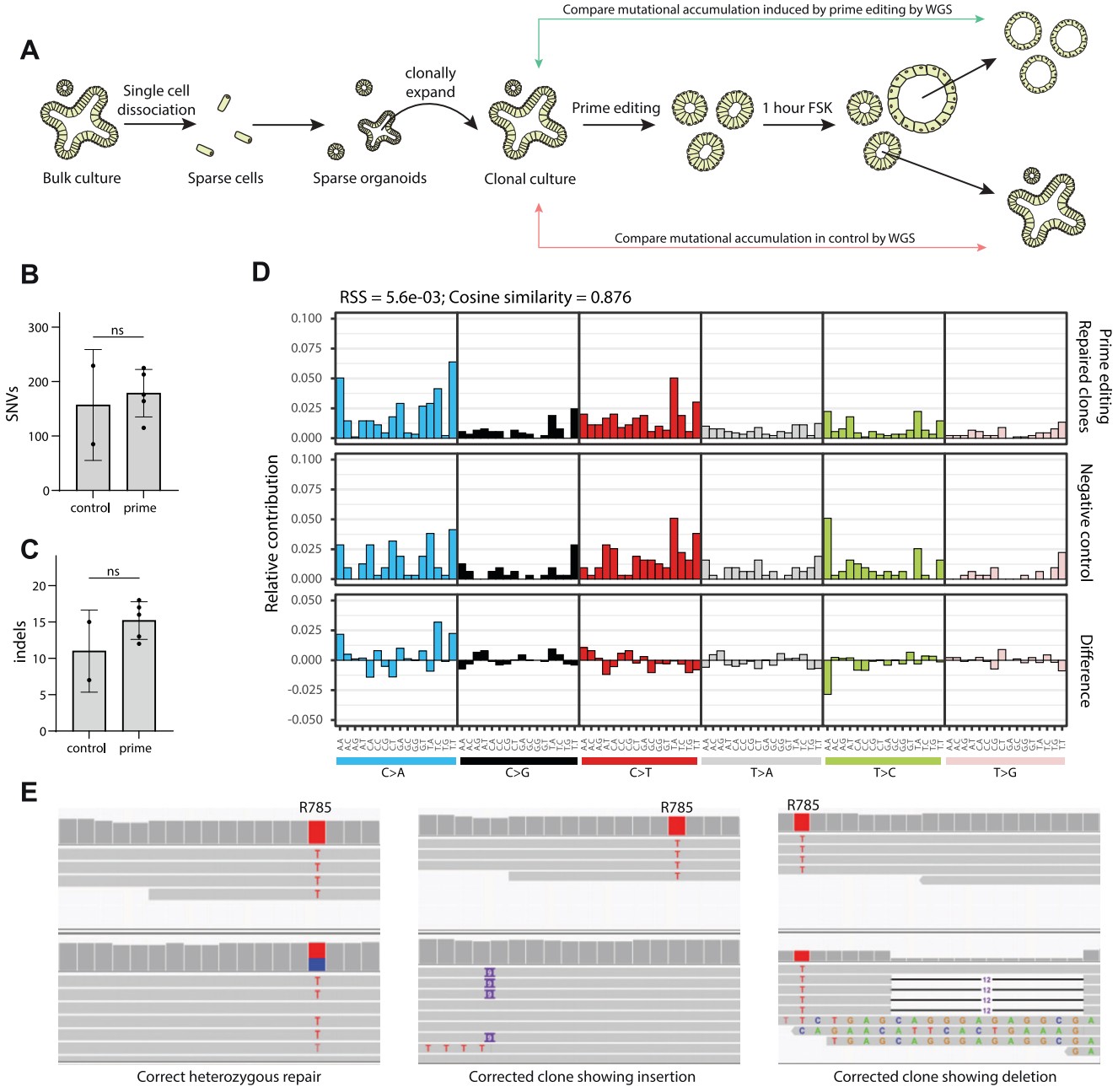

Figure 5. **Genome-wide off-target analysis of prime editing.**
**(A)** Schematic overview of the strategy to determine genome-wide off-target effects of prime editing. **(B)** Total amount of genome-wide single-nucleotide variant's as determined by whole-genome sequencing. **(C)** Total amount of genome-wide indels as determined by whole-genome sequencing. **(D)** Mutational signature analysis by relative contribution of context-dependent mutation types in two controls and five prime-edited clonal organoid lines. **(E)** Integrative Genomics Viewer representation of a correct heterozygous prime editing–mediated mutation repair, a clone harboring an insertion downstream of the target site, and a clone with a deletion upstream of the target site.

editing, prime editing may be a suitable technique. The most recent version of the CFTR2 database contains 442 mutations that have been described in CF patients (http://cftr2.org). Of these 442 mutations, 98 have a suitable PAM (Either NGG for SpCas9 or NGN for xCas9 and SpCas9-NG) for adenine base editing (Hu et al, 2018b; Nishimasu et al, 2018). A further 37 can be repaired by cytidine base editing. Thus, 30.5% of the mutations in *CFTR* can theoretically be repaired by base editors (Fig 4G). As prime editors are able to introduce DNA up to a size of 30 bp

into the genome at the target site, in principle 419 of 442 mutations (95%) can be repaired by this technique. This makes prime editing an interesting technique for *CFTR* repair.

## Prime editing does not result in genome-wide off-target effects

To explore the safety of prime editing in the repair of *CFTR*, we performed an off-target analysis by whole-genome sequencing

(WGS). We first generated a clonal line from our bulk *CFTR*-R785* colon organoid line, to avoid preexisting sequence heterogeneity in our organoid line. We transfected this clonal organoid line with pegRNA/PE3-guide pairs and the PE2 plasmids. We then picked five repaired organoid lines as indicated by FIS, 2 wk after transfection. As a control, we picked two non-repaired organoids. All lines were expanded for 2 wk to generate a sufficient amount for WGS (Fig 5A). WGS revealed no significant genome-wide differences in either single-nucleotide variants (SNV's) (Fig 5B) or indels (Fig 5C). Observed SNV's were uniformly scattered across all chromosomes, without bias towards any specific genomic region (Fig S5) As the sample size in our study was small and differences in organoids culturing and propagation of individual clones are difficult to control for, we used a mutational signature analysis (Alexandrov et al, 2013) to study base changes that could have been caused independent of cognate sgRNA binding of either the pegRNA or the PE3-guide. Mutational signature analysis did not show a difference in mutational patterns supporting the safety of prime editing (Cosine similarity = 0.876) (Fig 5D). Interestingly, even though we selected organoid for WGS by FIS responsiveness, we again observed indels around the target site in repaired organoid lines. Prime editing resulted in correct, heterozygous repair in three of five lines, but the other two clones carried a heterozygous 9-bp insertion downstream of the R785* mutation and a heterozygous 13-bp deletion directly upstream of the R785* mutation, respectively (Fig 5E). Overall, these results indicated that prime editing does not induce genome-wide off-target changes. However, performing Sanger sequencing around the target area remains key for determining correct mutational repair.

## Discussion

In this study, we first explore the use of prime editing for the modeling of oncogenic mutations in both hepatocyte and intestinal organoids. Our data imply that prime-editing efficiencies differ greatly between organoid tissue types. Anzalone et al (2019) described similar results in the original description of prime editing where efficiencies differed greatly between cell-lines (Anzalone et al, 2019). These differences between organs are important to keep in mind when designing disease modeling studies in vitro and in vivo. Moreover, we tested a total of 10 different target sequences (8 in *TP53* and 2 in *APC*). Of those 10 targets, only 4 resulted in correct modeling of the mutation in organoids with varying efficiency. This varying efficiency of prime editing has been shown previously in organoids (Schene et al, 2020) and in a wide variety of targets in HEK293T cells (Kim et al, 2021) and is striking as SpCas9 in general exhibits robust editing over all targets harboring a suitable NGG PAM (Kim et al, 2020). Further development of prime editing could potentially resolve these varying editing efficiencies and might increase the robustness of the technique.

Even though correct integration of the desired edits was achieved on a variety of targets, we also uncovered undesired edits, as has been seen before in mice (Aida et al, 2020 *Preprint*). Un-intended indel formation around the target site was often seen on one allele, and sometimes even on both alleles. This may be

explained by the need to generate a second nick on the opposing strand close to the initial nick by the PE2 machinery. The use of two sgRNAs that nick opposing strands is known to generate indels and is even often used to increase specificity of CRISPR/Cas9–mediated genome engineering (Ran et al, 2013). Further optimization of the prime-editing fusion protein may aim to render the generation of a second nick unnecessary and might therefore decrease unintended indel formation.

Over the past years, base editing plasmids have undergone several rounds of optimization turning them into efficient genome editors (Koblan et al, 2018; Zafra et al, 2018). Recently, similar efforts are being undertaken to increase effectivity of prime editing. NLS optimization of the PE2 fusion protein has been shown to increase editing efficiencies in adult mice in vivo (Liu et al, 2021). Moreover, the use of two pegRNA's in *trans* has been shown to increase prime-editing efficiency in plants (Lin et al, 2021). Utilization of these strategies might increase the effectivity of prime editing in human cell models. Finally, as has been previously shown by Schene et al (2020) prime editing does not introduce unwanted genome-wide off-target effects in the repair of *CFTR* and thus seems a safe strategy for gene repair.

In our hands, base editors are superior, both in terms of efficiency and of specificity in generating only the desired mutation (Geurts et al, 2020). However, if the desired edit cannot be generated by a base editor, for instance, if it regards an indel or non-transition base change, prime editing is a valuable alternative to CRISPR/Cas9–mediated HDR. Thus, prime editing is a versatile tool that can be used for disease modeling and clinical repair of most types of disease-causing mutations in human adult stem cells. Yet, it will require further improvement to allow widespread use as a technique for mutational modeling and for gene repair.

## Materials and Methods

### Organoid culture

Intestinal organoids are cultured as previously described (Sato et al, 2011). In short, the wild-type human colon organoid line P26n, as previously described in Van De Wetering et al (2015) was cultured in domes of Cultrex Pathclear Reduced Growth Factor Basement Membrane Extract (BME) (3533-001; Amsbio). Domes were covered by medium containing Advanced DMEM/F12 (Gibco), 1× Glutamax, 10 mmol/l Hepes, 100 μU/ml penicillin–streptomycin and 1× B27 (All supplied by Thermo Fisher Scientific), 1.25 mM N-acetylcysteine, 10 μM nicotinamide, 10 μM p38 inhibitor SB202190 (supplied by Sigma-Aldrich). This medium was supplemented with the following growth factors: 0.4 nM Wnt surrogate-Fc Fusion protein, 2% Noggin conditioned medium (U-Protein express), 20% Rspo1 conditioned medium (in-house), 50 ng ml EGF (Peprotech), 0.5 μM A83-01, and 1 μM PGE2 (Tocris). Intestinal organoids derived from people with CF are part of a large biobank at Hub for organoids (HUB), are stored in liquid nitrogen, and are passaged at least four times before electroporation experiments. CF organoids are kept in Matrigel (Corning) instead of BME. Furthermore, 2% Noggin-conditioned medium (U-protein express) is replaced by 10% Noggin conditioned medium (in-house). Moreover,

PGE2 is excluded and 30 $\mu$M of P38 inhibitor SB202190 (Sigma-Aldrich) is added to expansion medium for CF organoids. All organoids were passaged and split once a week 1:6 and filtered through a 40-$\mu$m cell strainer (Thermo Fisher Scientific) to remove differentiated structures from the culture. Hepatocyte organoids were cultured as previously described (Hu et al, 2018a).

## Plasmid construction

Human codon optimized prime-editing constructs were a kind gift from David Liu; pCMV_PE2_P2A_GFP (plasmid #132776; Addgene), pU6-pegRNA-GG-acceptor (plasmid #132777; Addgene). Human codon-optimized base editing constructs were a kind gift from David Liu; pCMV_ABEmax_P2A_GFP (plasmid #112101; Addgene). The empty sgRNA plasmid backbone was a kind gift from Keith Joung (BPK1520, plasmid #65777; Addgene). The SpCas9-expressing vector was created by using Q5 high-fidelity polymerase (NEB) to PCR-amplify the Cas9-P2A-GFP cassette from pSpCas9 (BB)-2A-GFP (PX458), a kind gift from Feng Zhang (plasmid #48138; Addgene). This Cas9-P2A-GFP cassette was then cloned into the PE2 expression vector NEBbuilder HIFI assembly master mix according to the manufacturer's protocols (NEB). pegRNA was created as previously described (Anzalone et al, 2019). In brief, the pU6-pegRNA-GG-acceptor plasmid was digested overnight using BsaI-HFv2 (NEB), loaded on a gel, and the 2.2-kb band was extracted using the QIAquick Gel extraction kit. Oligonucleotide duplexes for the spacer, scaffold and 3'-extension with their appropriate overhangs were annealed and cloned into the digested pUF-pegRNA-GG-acceptor by golden gate assembly according to the previously described protocol (Anzalone et al, 2019). PE3-guides and guides for both base editing and HDR experiments were cloned using inverse PCR together using BPK1520 as template and Q5 high-fidelity polymerase. Upon PCR cleanup (Qiaquick PCR purification kit), amplicons were ligated using T4 ligase and Dpn1 (both NEB) to get rid of template DNA. All transformations in this study were performed using OneShot Mach1t1 (Thermo Fisher Scientific) cells and plasmid identity was checked by Sanger sequencing (Macrogen). All constructed guide-RNA sequences can be found in Table S1.

## Organoid electroporation

Organoid electroporation was performed with slight modifications to this previously described protocol (Fujii et al, 2015; Geurts et al, 2020). Wild-type colon and intestinal organoids derived from CF patients were maintained in their respective expansion medium up until 2 d before electroporation. 2 d in advance, the expansion medium was switched to electroporation medium which does not contain the growth factors wnt and Rspo1. R-spondin–conditioned medium was replaced by Advanced DMEM-F12 (Gibco) supplemented by 1× Glutamax, 10 mmol/l Hepes, and 100 $\mu$U/ml penicillin–streptomycin (Thermo Fisher Scientific). Furthermore, the GSK-3 inhibitor CHIR99021 (Sigma-Aldrich) was added to the medium for wnt pathway activation and rho-kinase inhibitor Y-27632 (AbMole BioScience) was added to inhibit anoikis. 1 d before electroporation, 1.25% (vol/vol) DMSO was added to the organoid medium. On the day of electroporation, the organoids were dissociated into single cells using TrypLE (Gibco) supplemented with Y-27632 at 37°C for 15 min.

During the single-cell dissociation, the organoid suspension was vigorously pipetted every 5 min to keep the solution homogenous. $10^6$ cells per electroporation were resuspended in BTXpress solution and combined with 10 $\mu$l plasmid solution containing 7.5 $\mu$g pCMV_PE2_-P2A_GFP, pCMV_SpCas9, or pCMV_ABEmax_P2A_GFP depending on gene editing strategy and 2.5 $\mu$g per guide-RNA plasmid. In HDR experiments, 2.5 $\mu$g of single-stranded donor oligonucleotide, containing a WT-*CFTR* sequence and silent mutations to block Cas9 cleavage after repair was added to the plasmid mix. Electroporation was performed using NEPA21 with settings described before (Fujii et al, 2015). After electroporation, the cells were resuspended in 600 $\mu$l Matrigel or BME (50% Matrigel/BME and 50% expansion medium) and plated out in 20 $\mu$l droplet/well of a pre-warmed 48-well tissue culture plate (Greiner). After polymerization, the droplets were immersed in 300 $\mu$l of expansion medium and the organoids were maintained at 37°C and 5% CO2.

## Phenotypic selection by FIS

After electroporation, organoids were expanded for 7 d and subsequently replated in 72 wells of 48-well tissue culture plates (Greiner) to make organoids sufficiently sparse. Selection of genetically corrected organoids was based on CFTR function restoration as assessed by adding foskolin (5 $\mu$M) to the expansion medium. Pictures were made (1.25× on an EVOS FL Auto Imaging system) before and 60 min after forskolin addition. Organoids that showed swelling after 60 min were individually picked with a p200 pipette and a bend p200 pipette tip. Each individual genetically corrected organoid was dissociated into single cells using TrypLE supplemented with Y-27632 (10 $\mu$M) for 10 min at 37°C. The cells were plated in 20 $\mu$l Matrigel droplets/picked organoid (50% Matrigel and 50% CCM+) in prewarm 48-well tissue culture plates (Greiner) and maintained at 37°C and 5% CO$_2$.

## Phenotypic selection for oncogenic mutations

After electroporation, organoids were expanded for 5 d to offer sufficient time for recovery of the transfected cells. In prime-editing experiments with the goal to mutate *TP53*, 10 $\mu$M Nutlin-3 was added to the expansion medium. In prime-editing experiments with the goal to mutate APC, both wnt surrogate and Rspo1 were removed from the expansion medium. After 2 wk, individual organoids that survived selection were manually picked and clonally expanded as previously described.

## Genotyping of clonal organoid lines

Organoid DNA was harvested from 10 to 20 $\mu$l Matrigel/BME suspension and DNA was extracted using the Zymogen Quick-DNA Microprep kit. Target regions were amplified from the genome using Q5 high-fidelity polymerase using primers. Sequencing was performed using the M13F tail as all forward amplification primers for targeted sequencing contained a tail with this sequence. Prime editing, base editing, and CRISPR/Cas9–mediated HDR induced genomic alterations were confirmed by Sanger sequencing (Macrogen). SubsequentSsanger trace deconvolution was performed with the use of the online tool ICE by Synthego. Primers used for PCR amplification and sequencing can be found in Table S2.

### FIS-assay

To quantify CFTR function in the genetically corrected intestinal organoids, we conducted the FIS-assay. This was performed in duplicates at three independent culture time points (n = 3) according to previously published protocols (Boj et al, 2017; Vonk et al, 2020). In brief, intestinal organoids were seeded in 96-well culture plates in 4 μl of 50% Matrigel. Each Matrigel dome contained roughly 20–40 organoids and was immersed in expansion medium. The day after, organoids were incubated for 30 min with 3 μM calcein green (Invitrogen) to fluorescently label the organoids and stimulated with 5 μM forskolin. Every 10 minutes, the total calcein green–labeled area per well was monitored by a Zeiss LSM800 confocal microscope, for 60 min while the environment was maintained at 37°C and 5% $CO_2$. A Zen Image analysis software module (Zeiss) was used to quantify the organoid response (area under the curve measurements of relative size increase in organoids after 60 min forskolin stimulation, t = 0 min baseline of 100%).

### Efficiency calculation of prime editing in organoids

pegRNA/PE3-guide-RNA pairs were co-transfected with 10 μg PiggyBac transposon system (2.8 μg transposase + 7.2 μg hygromycin resistance containing transposon [Andersson-Rolf et al, 2017]) as described before using the NEPA21. 5 d post transfection organoid culture medium was supplemented with 100 μg/μl Hygromycin B gold (InvivoGen). 14 d after selection of clonal organoids, surviving hygromycin selection, were individually picked and Sanger sequencing was performed as previously described. Primer sequences can be found in Table S2. For comparison of prime editing to conventional CRISPR/Cas9–mediated HDR and adenine base editing on the *CFTR*-R785* mutation, we transfected pegRNA/PE3-guide pairs with pCMV_PE2_P2A_GFP and pCMV_SpCas9 with the respective sgRNA/HDR repair template combination (ssDNA oligo Table S1) and pCMV_ABEmax_P2A_GFP with respective repair sgRNA were transfected in duplicates as previously described. 3 d post transfection, organoids were dissociated to single cells using TrypLE (Gibco) supplemented with Y-27632 at 37°C for 15 min. During the single-cell dissociation, the organoid suspension was vigorously pipetted every 5 min to keep the solution homogenous. Single-cell suspensions were filtered and GFP positive, and thus transfected cells were sorted and plated at a concentration of 500 cells per 100 μl using a FACSMelody (BD Biosciences). We then quantified the total amount of cultured organoids by using cellprofiler 3.1.5. Editing efficiencies were determined by dividing the total amount of organoids by the transfection efficiency and the amount of FIS assay–responsive organoids 2 wk after plating.

### Whole-genome sequencing and mapping

Genomic DNA was isolated from 100 μl of Matrigel/organoid suspension using DNeasy Blood & Tissue Kit, according to the protocol. Standard Illumina protocols were applied to generate DNA libraries for Illumina sequencing from 20 to 50 ng of genomic DNA. All samples (five genetically corrected clones, two non-corrected control samples of the R785X/R785X donor, and the clonal line before prime editing) were sequenced (2 × 150 bp) by using Illumina NovaSeq to 15× base coverage. Reads were mapped against human

reference genome GRCh38 using Burrows-Wheeler Aligner v0.7.17 (Li & Durbin, 2010), with settings "bwa mem -c 100 -M." Duplicate sequence reads were marked using Sambamba v0.7.0 and recalibrated using the GATK BaseRecalibrator v4.1.3.0. More details on the pipeline can be found on https://github.com/ToolsVanBox/NF-IAP.

### Mutation calling and filtering

Raw variants were multisample-called by using the GATK HaplotypeCaller v4.1.3.0 (Depristo et al, 2011). The quality of variant and reference positions was evaluated by using GATK VariantFiltration v4.1.3.0 with options "QD < 2.0" –filter-expression "mapping quality (MQ) < 40.0" –filter-expression "FS > 60.0" –filter-expression "HaplotypeScore > 13.0" –filter-expression "MQRankSum < –12.5" –filter-expression "ReadPosRankSum < –8.0" –filter-expression "MQ0 ≥ 4 && ((MQ0/(1.0 × DP)) > 0.1)" –filter-expression "DP < 5" –filter-expression "QUAL < 30" –filter-expression "QUAL ≥ 30.0 && QUAL < 50.0" –filter-expression "SOR > 4.0" –filter-name "single nucleotide polymorphism (SNP)_LowQualityDepth" –filter-name "SNP_MappingQuality" –filter-name "SNP_StrandBias" –filter-name "SNP_HaplotypeScoreHigh" –filter-name "SNP_MQRankSumLow" –filter-name "SNP_ReadPosRankSumLow" –filter-name "SNP_HardToValidate" –filter-name "SNP_LowCoverage" –filter-name "SNP_VeryLowQual" –filter-name "SNP_LowQual" –filter-name "SNP_SOR" -cluster 3 -window 10. To obtain high-quality somatic mutation catalogs, we applied post processing filters as described (Blokzijl et al, 2016). Briefly, we considered variants at autosomal chromosomes without any evidence from a paired control sample (a clone before editing); passed by VariantFiltration with a GATK phred-scaled quality score ≥250; a base coverage of at least 10× in the clonal and paired control sample; no overlap with SNPs in the Single Nucleotide Polymorphism Database v146; and absence of the variant in a panel of unmatched normal human genomes (BED-file available upon request). We additionally filtered base substitutions with a GATK genotype score (GQ) lower than 99 or 10 in the clonal or paired control sample, respectively. For indels, we filtered variants with a GQ score lower than 99 in both the clonal and paired control sample and filtered indels that were present within 100 bp of a called variant in the control sample. In addition, for both SNVs and INDELs, we only considered variants with a MQ score of 60 and with a variant allele frequency of 0.3 or higher in the clones to exclude in vitro accumulated mutations (Blokzijl et al, 2018; Jager et al, 2018). The scripts used are available on GitHub (https://github.com/ToolsVanBox/SMuRF). The distribution of variants was visualized using an in-house–developed R package (MutationalPatterns) (Blokzijl et al, 2018).

### Mutational signature analysis

We extracted mutational signatures and estimated their contribution to the overall mutational profile as described using an in-house–developed R package (MutationalPatterns) (Blokzijl et al, 2018). In this analysis, we included small intestine data (previously analyzed) to explicitly extract in vivo and in vitro accumulated signatures (Blokzijl et al, 2016). To determine the transcriptional strand contribution and bias, we selected all point mutations that fall within gene bodies and checked whether the mutated base was located on the transcribed or non-transcribed strand. We used a in house

developed R package (MutationalPatterns) to determine transcriptional strand bias as described (Blokzijl et al, 2018).

## Data Availability

The whole-genome sequencing data from this publication have been deposited to the European Genome-phenome Archive (https://ega-archive.org/) and assigned the identifier: EGAS00001005358. All software tools used for sequencing data analysis can be found online at: https://github.com/ToolsVanBox.

## Supplementary Information

## Acknowledgements

This work was supported by Cancer Research UK C6307/A29058 and The Mark Foundation for Cancer Research (H Clevers and MH Geurts). This work was supported by the NWO building blocks of life project: Cell dynamics within lung and intestinal organoids (737.016.009) (MH Geurts) and grants of the Dutch Cystic Fibrosis Foundation (NCFS), The Netherlands, as part of the HIT-CF Program; the Dutch Health Organization ZonMw, The Netherlands. This work was supported by the Netherlands Organ-on-Chip Initiative, an NWO Gravitation project (024.003.001) funded by the Ministry of Education, Culture and Science of the government of The Netherlands (H Clevers, C Pleguezuelos-Manzano). This work is part of the Oncode Institute which is partly financed by the Dutch Cancer Society and was funded by the gravitation program CancerGenomiCs.nl from The Netherlands Organisation for Scientific Research (NWO). M Boretto and A Andersson-Rolf are postdoctoral researchers supported by a long-term fellowship of the European Organization for Molecular Biology (EMBO/ALTF 765-2019, and EMBO ALTF 332-2018, resp.).

### Author Contributions

MH Geurts: conceptualization, data curation, formal analysis, validation, investigation, visualization, methodology, project administration, and writing—original draft, review, and editing.
E de Poel: data curation, formal analysis, validation, investigation, visualization, and writing—review and editing.
C Pleguezuelos-Manzano: formal analysis, investigation, methodology, and writing—review and editing.
R Oka: software, visualization, and writing—review and editing.
L Carrillo: validation, investigation, methodology, and writing—review and editing.
A Andersson-Rolf: supervision, methodology, and writing—review and editing.
M Boretto: investigation, methodology, and writing—review and editing.
JE Brunsveld: software and visualization.
R van Boxtel: software, supervision, and writing—review and editing.
JM Beekman: supervision and writing—review and editing.
HC Clevers: conceptualization, supervision, funding acquisition, and writing—original draft, review, and editing.

### Conflict of Interest Statement

JM Beekman is an inventor on (a) patent(s) related to the FIS assay and received financial royalties from 2017 onward. JM Beekman reports receiving (a) research grant(s) and consultancy fees from various industries, including Vertex Pharmaceuticals, Proteostasis Therapeutics, Eloxx Pharmaceuticals, Teva Pharmaceutical Industries, and Galapagos outside the submitted work. H Clevers holds several patents on organoid technology. Their application numbers are as follows: PCT/NL2008/050543, WO2009/022907; PCT/NL2010/000017, WO2010/090513; PCT/IB2011/002167, WO2012/014076; PCT/IB2012/052950, WO2012/168930; PCT/EP2015/060815, WO2015/173425; PCT/EP2015/077990, WO2016/083613; PCT/EP2015/077988, WO2016/083612; PCT/EP2017/054797, WO2017/149025; PCT/EP2017/065101, WO2017/220586; PCT/EP2018/086716; and GB1819224.5.

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
