## [Reviewer comments · Life Science Alliance]

Life Science Alliance

Evaluating CRISPR-based Prime Editing for cancer modeling and CFTR repair in organoids

Maarten Geurts, Eyleen de Poel, Cayetano Pleguezuelos-Manzano, Rurika Oka, Léo Carrillo, Amanda Andersson-Rolf, Matteo Boretto, Jesse Brunsveld, Ruben van Boxtel, Jeffrey Beekman, and Hans Clevers

DOI: <https://doi.org/10.26508/lsa.202000940>

Corresponding author(s): Hans Clevers, Hubrecht Institute

Review Timeline:	Submission Date:	2020-10-19
	Editorial Decision:	2020-11-11
	Revision Received:	2021-06-01
	Editorial Decision:	2021-06-25
	Revision Received:	2021-07-13
	Accepted:	2021-07-16

Transaction Report:

November 11, 2020

Re: Life Science Alliance manuscript #LSA-2020-00940-T

Prof. Hans C. Clevers
Hubrecht Institute
Clevers group
Uppsalalaan 8
Utrecht, Utrecht 3584CT
Netherlands

Dear Dr. Clevers,

Thank you for submitting your manuscript entitled "Evaluating CRISPR-based Prime Editing for cancer modeling and CFTR repair in intestinal organoids" to Life Science Alliance (LSA). The manuscript was assessed by expert reviewers, whose comments are appended to this letter.

As you will note from the reviewer comments below, the reviewers were intrigued by these findings, but do raise some technical issues that need to be addressed. All the concerns that the reviewers have raised need to be addressed, prior to further consideration of the manuscript at LSA. Since this is a little more extensive than the usual revisions required for LSA, we would appreciate if you can get back to us with a rudimentary plan as to how you will address these concerns, and a confirmation that you would be interested in submitting a revision to us.

Thank you for this interesting contribution to Life Science Alliance. We are looking forward to receiving your revised manuscript.

Sincerely,

Shachi Bhatt, Ph.D.
Executive Editor
Life Science Alliance
<https://www.lsa-journal.org/>
Tweet @SciBhatt @LSAJournal

- A letter addressing the reviewers' comments point by point.
- An editable version of the final text (.DOC or .DOCX) is needed for copyediting (no PDFs).
- High-resolution figure, supplementary figure and video files uploaded as individual files: See our detailed guidelines for preparing your production-ready images, <https://www.life-science-alliance.org/authors>
- Summary blurb (enter in submission system): A short text summarizing in a single sentence the study (max. 200 characters including spaces). This text is used in conjunction with the titles of papers, hence should be informative and complementary to the title and running title. It should describe the context and significance of the findings for a general readership; it should be written in the present tense and refer to the work in the third person. Author names should not be mentioned.

B. MANUSCRIPT ORGANIZATION AND FORMATTING:

Reviewer #1 (Comments to the Authors (Required)):

This manuscript by Geurts et al. evaluates the use of prime editing to generate cancer cell line models and repair CFTR mutations in adult human stem cell-derived intestinal organoids. In the first half of the manuscript, authors attempted to use PE3 to generate clonal organoid cell lines representing seven common TP53 pathogenic mutations and two APC pathogenic mutations. A single pegRNA and nicking sgRNA combination was evaluated for the installation of each desired

mutation and edited organoids were isolated under culture conditions designed to select clonal pathogenic mutants of each respective gene. Three of the seven desired TP53 mutant organoid cell lines were identified and one APC mutant was identified, although the identified APC mutant had an indel in place of the desired stop codon edit. The authors proceeded to evaluate prime editing for the correction pathogenic CFTR mutations using organoids from cystic fibrosis patient-derived intestinal stem cells. Using eight combinations of pegRNAs and nicking sgRNAs, the authors attempted to correct the F508del mutation of CFTR with PE3 and compared this strategy to CRISPR/Cas9-mediated HDR. Using a forskolin induced swelling (FIS) response to identify clonal organoids with repaired CFTR genes, the authors identified 108 clonal organoids repaired by their HDR-based CFTR F508del correction strategy and 3 clonal organoids repaired by their best PE3-based CFTR F508del correction strategy. Similarly, the authors attempted to correct the R785X CFTR mutation with PE3 and HDR but also included an adenine base editing (ABE) strategy. Based on their FIS assay of edited CFTR R785X organoids they found 307 ABE-corrected clonal organoids, 42 HDR-corrected clonal organoids, and 55 PE3-correct clonal organoids. For both F508del and R785X correction experiments 1-3 HDR, ABE or PE3-treated clones total were evaluated for the presence of indels by Sanger sequencing and some indels were identified. From the experiments to correct the CFTR F508del and R785X mutations the authors concluded prime editing efficiencies were generally low and comparable to HDR, and that ABE - as a more mature technology - remains superior to prime editing in terms of editing efficiency and of specificity.

The results presented in this manuscript are a confirmation of the recently published organoid editing results by Schene et al. in Nature Communications in 2020 and the original prime editing results from Anzalone et al. 2019 in Nature. The TP53 data supports the conclusion that useful model cell lines can be generated by prime editing in adult human stem cell-derived intestinal organoids. However, the authors' assertions about the efficiency and safety of prime editing in comparison to CRISPR/Cas9 HDR and adenine base editing are not convincing and require substantial additional experimentation. The major claim that prime editing can be used to generate useful edits in this unique model system is significant, although confirmatory, and the major claim about the comparative editing efficiency and precision between ABE, HDR, and PE is also confirmatory, although poorly supported. This work should be of interest to those specifically interested in editing adult human stem cell-derived intestinal organoids, in particular at the genes presented. If the major points listed below can be addressed in a satisfactory manner, this manuscript would be suitable for publication.

Major Points:

1. Throughout the manuscript editing results are not quantified as a percent of the total population of treated cells but rather as the number of organoid clones counted following a selection or phenotypic screen. This is somewhat suitable for assessing the capability of prime editing to generate clonal model cell lines with specific pathogenic mutations (as described for the TP53 and APC experiments) but is not useful for assessing the efficiency of a corrective therapeutic edit (as described for the correction of the CFTR F508del and R785X mutations). This choice of quantification makes it difficult to draw a comparison between the actual editing efficiencies of the methods presented (PE3, HDR, ABE) and does not support the authors' conclusions about the efficiency prime editing in comparison to HDR or ABE as written. In Geurts et al. 2020 in Cell Stem Cell, the same group presented methods to quantify editing as a percent of total treated cells by relying on an ABE plasmid with a P2A GFP reporter and a FIS response assay, specifically at R785X, which is an editing target also described in this manuscript. It would benefit readers immensely to have a similar quantification of editing to facilitate a comparison of the genome editing methods presented.

2. Perhaps most important, the authors acknowledge the importance of trying multiple combinations of pegRNA PBS and RT template lengths, yet only try 4 combinations - far too few - of pegRNAs and nicking sgRNAs for their direct comparison of PE and ABE at CFTR R785X. From this single comparison, they conclude that ABE is superior in terms of editing efficiency. The authors do not convincingly support this conclusion in their experiments. Anzalone et al. 2019 and subsequent prime editing papers all underscore the importance of trying multiple pegRNA PBS and RT lengths, and also testing multiple nicking sgRNAs. For the R785X mutation presented, the authors only present two PBS lengths and one RT template length for each pegRNA tested and only test one nicking sgRNA for each pegRNA tested. From the guidance offered in Anzalone et al. 2019 these tested conditions are far too narrow. Additionally, to make the conclusion that ABE is superior to PE in terms of editing efficiency, the authors only perform a direct comparison of PE3 and ABE editing at one target. The scope of this comparison is too narrow to support this conclusion and the authors need to better support their claims by conducting similar comparisons at other genomic targets, preferable in a diversity of genes. Additionally, the experiments to generate the TP53 and APC clonal organoid models relied only on single pegRNA and nicking sgRNA combinations to create each desired mutant, with only three of seven TP53 mutants and no APC mutants being generated. At a minimum, many more (several dozen) additional pegRNA and nicking sgRNA designs and combinations should be evaluated for creation of a variety of mutants before such conclusions are even preliminarily supported. The original attempts with such a narrow selection of prime editing reagents fall outside of the guidance described in Anzalone et al. 2019 and suggest that it is not possible to generate the desired clonal organoid models with prime editing.

3. The authors conclude that ABE is superior prime editing in terms of safety based on Sanger sequencing results used to identify on-target indels from a handful of clones from their CFTR R785X experiments. Presently, assessing the "safety" of genome editing methods outside of a clinical setting is understood to mean assessing the frequency of unintended off-target editing events using in vitro or cell-based methods. The best approaches for assessing this aspect of safety are unbiased genome-wide off-target detection methods - for PE3 these methods could include the modified nDigenome-seq approach recently described by Kim et al. 2020 in *Nucleic Acids Research* or the WGS approach described in Geurts et al. 2020 in *Cell Stem Cell*. For the CFTR R785X experiment described "safety" appears to describe the frequency of on-target indels induced by PE3, HDR, or ABE. Using the described method of picking clonal organoids and Sanger sequencing, it would take Sanger sequencing results collected from hundreds of clones to make a convincing comparison of indel frequencies between PE3, HDR, and ABE. Alternatively, a high-throughput sequencing approach to quantify both correct editing and indels from pooled gDNA of all electroporated clones would provide sufficient evidence for the authors to compare editing efficiencies and indel frequency between PE3, HDR, and ABE in adult human stem cell-derived intestinal organoids.

Minor Points

1. Figures 1 and 2 should be combined, TP53 data is split across both figures and makes it difficult to follow the author's narrative.

Reviewer #2 (Comments to the Authors (Required)):

1. A short summary of the paper, including description of the advance offered to the field. The current manuscript by Geurts and colleagues describes the application of a recently reported

gene editing technology called prime editing to repair CFTR mutations. The new technology is discussed in comparison to the groups prior data using other CFRSPR editing technologies however do not directly show comparisons of the data in the manuscript. The manuscript reports on efficiency and off target undesired effects of prime editing using examples of insertion and correction of mutations. The manuscript provides a proof of principle study detailing prime editing as an editing tool with wider application than prior tested techniques. While the method may have potential clinical relevance, the technological advance is incremental and associated with inefficiencies and off target effects associated with the prime editing technology. These comparative observations, however, could be important to the field when considering use of the new technologies.

2. For each main point of the paper, please indicate if the data are strongly supportive. If not, explicitly state the additional experiments essential to support the claims made and the timeframe that these would require.

The manuscript describes and compares the use of a new gene editing technology - called prime editing - for the functional repair of mutations in human intestinal cells. They apply the technology to 4 genes: 1) TP53, 2) PE3, 3) CFTR F508 and 4) CFTR R785.

The main take home messages are that

1) prime editing can be used to generate oncogenic mutations in intestinal organoids
a. The efficiencies reported in this manuscript seem to be dependent upon one experimental repeat in one biological specimen for each mutation and reports on the number of clones generated. This gives no indication of the efficiency - in the absence of editing and selection how many clones are usually observed? Is it possible that these mutations lead to complications with self-renewal and proliferation resulting in fewer clones?

2) prime editing can be used for the functional repair of mutations causative for cystic fibrosis
a. Similarly, for the CFTR organoids the data seems to be from one patient's cells and an experimental repeat of 1 for each combination of reagents (Panel 3D). Given high patient to patient variability in these procedures it would be nice to see biological and experimental repeats to truly gauge the efficiency of the technique

3) Base editors are confirmed to be superior in terms of efficiency and specificity however are not able to target substantial regions of the genome. Prime editing is more versatile but is associated with lower efficiency and insertion of undesired edits on one or both alleles. It would be nice to have direct comparisons shown in the manuscript - perhaps the addition of a table would be very beneficial for the reader to compare the technologies.

3. Lastly, indicate any additional issues you feel should be addressed (text changes, data presentation, statistics etc.)

The manuscript perhaps overstates the application of the new technology in the results sections based on the findings of the manuscript. It would be good to provide a more in-depth discussion of the technology's merits and issues in the discussion

To:
Shachi Bhatt, PhD.
Executive Editor
Life Science Alliance

Dear Shachi,

We respectfully resubmit our manuscript LSA-2020-00940-T entitled: "Evaluating CRISPR-based Prime Editing for cancer modeling and CFTR repair in organoids" by Maarten H. Geurts et al. We thank the reviewers for their very useful comments and have edited the manuscript to address their points.

An overview of our additions and improvements:

- We have included comparisons between hepatocyte and intestinal organoids to further explore prime editing in an independent organoid model.
- To further substantiate our findings that base editing outperforms prime editing we have added an additional target in TP53, Y220C where we directly compare both techniques.
- We have added an additional organoid line derived from a cystic fibrosis patient harboring the CFTR-F508del mutation
- We have included additional pegRNA/PE3-guide pairs and included an extensive efficiency experiment in the comparison of prime- and base editing mediated repair of the CFTR-R785* mutation
- Whole genome sequencing has been performed to underscore the lack of off-target effects induced by prime editors.

We hope that -with these improvements- Life Science Alliance can now publish our manuscript.

Kind regards, on behalf of all authors
Hans Clevers

Reviewer #1 (Comments to the Authors (Required)):

This manuscript by Geurts et al. evaluates the use of prime editing to generate cancer cell line models and repair CFTR mutations in adult human stem cell-derived intestinal organoids. In the first half of the manuscript, authors attempted to use PE3 to generate clonal organoid cell lines representing seven common TP53 pathogenic mutations and two APC pathogenic mutations. A single pegRNA and nicking sgRNA combination was evaluated for the installation of each desired mutation and edited organoids were isolated under culture conditions designed to select clonal pathogenic mutants of each respective gene. Three of the seven desired TP53 mutant organoid cell lines were identified and one APC mutant was identified, although the identified APC mutant had an indel in place of the desired stop codon edit. The authors proceeded to evaluate prime editing for the correction pathogenic CFTR mutations using organoids from cystic fibrosis patient-derived intestinal stem cells. Using eight combinations of pegRNAs and nicking sgRNAs, the authors attempted to correct the F508del mutation of CFTR with PE3 and compared this strategy to CRISPR/Cas9-mediated HDR. Using a forskolin induced swelling (FIS) response to identify clonal

organoids with repaired CFTR genes, the authors identified 108 clonal organoids repaired by their HDR-based CFTR F508del correction strategy and 3 clonal organoids repaired by their best PE3-based CFTR F508del correction strategy. Similarly, the authors attempted to correct the R785X CFTR mutation with PE3 and HDR but also included an adenine base editing (ABE) strategy. Based on their FIS assay of edited CFTR R785X organoids they found 307 ABE-corrected clonal organoids, 42 HDR-corrected clonal organoids, and 55 PE3-correct clonal organoids. For both F508del and R785X correction experiments 1-3 HDR, ABE or PE3-treated clones total were evaluated for the presence of indels by Sanger sequencing and some indels were identified. From the experiments to correct the CFTR F508del and R785X mutations the authors concluded prime editing efficiencies were generally low and comparable to HDR, and that ABE - as a more mature technology - remains superior to prime editing in terms of editing efficiency and of specificity.

The results presented in this manuscript are a confirmation of the recently published organoid editing results by Schene et al. in Nature Communications in 2020 and the original prime editing results from Anzalone et al. 2019 in Nature. The TP53 data supports the conclusion that useful model cell lines can be generated by prime editing in adult human stem cell-derived intestinal organoids. However, the authors' assertions about the efficiency and safety of prime editing in comparison to CRISPR/Cas9 HDR and adenine base editing are not convincing and require substantial additional experimentation. The major claim that prime editing can be used to generate useful edits in this unique model system is significant, although confirmatory, and the major claim about the comparative editing efficiency and precision between ABE, HDR, and PE is also confirmatory, although poorly supported. This work should be of interest to those specifically interested in editing adult human stem cell-derived intestinal organoids, in particular at the genes presented. If the major points listed below can be addressed in a satisfactory manner, this manuscript would be suitable for publication.

Major Points:

1. Throughout the manuscript editing results are not quantified as a percent of the total population of treated cells but rather as the number of organoid clones counted following a selection or phenotypic screen. This is somewhat suitable for assessing the capability of prime editing to generate clonal model cell lines with specific pathogenic mutations (as described for the TP53 and APC experiments) but is not useful for assessing the efficiency of a corrective therapeutic edit (as described for the correction of the CFTR F508del and R785X mutations). This choice of quantification makes it difficult to draw a comparison between the actual editing efficiencies of the methods presented (PE3, HDR, ABE) and does not support the authors' conclusions about the efficiency prime editing in comparison to HDR or ABE as written. In Geurts et al. 2020 in Cell Stem Cell, the same group presented methods to quantify editing as a percent of total treated cells by relying on an ABE plasmid with a P2A GFP reporter and a FIS response assay, specifically at R785X, which is an editing target also described in this manuscript. It would benefit readers immensely to have a similar quantification of editing to facilitate a comparison of the genome editing methods presented.

A: We have now included additional experiments in the manuscript, allowing efficiency calculations. For our experiments regarding modeling oncogenic mutations in TP53, we have co-transfected plasmids that contain genome editing components with PiggyBac plasmids that convey resistance towards Hygromycin. We then pick single clones that survive hygromycin selection (and have thus been transfected). We then performed sanger sequencing on 36 clones on in total 7 targets (R175H Liver versus intestine, R249S in liver VS intestine, Y220C base editing versus prime editing in intestine and C176F in intestine). Hereby we obtain a good insight into the editing efficiencies on a diverse set

of targets in two independent organs. We have also, as suggested, included a similar efficiency experiment for the CFTR-R785 mutation as previously described in Geurts et al. Cell Stem Cell 2020. Here we use fluorescence to sort out cells that have been transfected and we then perform the Forskolin induced swelling assay to score for (functional) gene correction.*

2. Perhaps most important, the authors acknowledge the importance of trying multiple combinations of pegRNA PBS and RT template lengths, yet only try 4 combinations - far too few - of pegRNAs and nicking sgRNAs for their direct comparison of PE and ABE at CFTR R785X. From this single comparison, they conclude that ABE is superior in terms of editing efficiency. The authors do not convincingly support this conclusion in their experiments. Anzalone et al. 2019 and subsequent prime editing papers all underscore the importance of trying multiple pegRNA PBS and RT lengths, and also testing multiple nicking sgRNAs. For the R785X mutation presented, the authors only present two PBS lengths and one RT template length for each pegRNA tested and only test one nicking sgRNA for each pegRNA tested. From the guidance offered in Anzalone et al. 2019 these tested conditions are far too narrow. Additionally, to make the conclusion that ABE is superior to PE in terms of editing efficiency, the authors only perform a direct comparison of PE3 and ABE editing at one target. The scope of this comparison is too narrow to support this conclusion and the authors need to better support their claims by conducting similar comparisons at other genomic targets, preferable in a diversity of genes. Additionally, the experiments to generate the TP53 and APC clonal organoid models relied only on single pegRNA and nicking sgRNA combinations to create each desired mutant, with only three of seven TP53 mutants and no APC mutants being generated. At a minimum, many more (several dozen) additional pegRNA and nicking sgRNA designs and combinations should be evaluated for creation of a variety of mutants before such conclusions are even preliminarily supported. The original attempts with such a narrow selection of prime editing reagents fall outside of the guidance described in Anzalone et al. 2019 and suggest that it is not possible to generate the desired clonal organoid models with prime editing.

A: We acknowledge that we do not use the same number of pegRNA/PE3-guide combinations as has been described in Anzalone et al 2019. However, in this original description of prime editing, the model of use is the 2D cell line HEK293T. Growth-speed and transfection efficiencies in this cell line far exceed those of our 3D organoid models (which essentially are comprised of primary cells), making it more suitable for high-throughput analysis on a large number of pegRNA/PE3-guide combinations. Nevertheless, we have now doubled the number of combinations from 4 to 8 in the comparison of base editing to prime editing on the repair of the CFTR-R785 mutation.*

Moreover, the superior editing efficiency of base editing over prime editing was described by Anzalone et al in the original description of prime editing. Our comparison of base- and prime editing on the repair of CFTR-R785 underscores this data. To further substantiate these findings we have included an additional target to compare both editing strategies, TP53-Y220C. In this experiment we again see increased efficiency of base editing. However, we also see editing on additional targets within the editing window of the base editor, as has been described by Anzalone et al. Our data thus underscores these original findings.*

3. The authors conclude that ABE is superior prime editing in terms of safety based on Sanger sequencing results used to identify on-target indels from a handful of clones from their CFTR R785X experiments. Presently, assessing the "safety" of genome editing methods outside of a clinical setting is understood to mean assessing the frequency of unintended off-target editing events using in vitro or cell-based methods. The best approaches for assessing this aspect of safety are unbiased

genome-wide off-target detection methods - for PE3 these methods could include the modified nDigenome-seq approach recently described by Kim et al. 2020 in Nucleic Acids Research or the WGS approach described in Geurts et al. 2020 in Cell Stem Cell. For the CFTR R785X experiment described "safety" appears to describe the frequency of on-target indels induced by PE3, HDR, or ABE. Using the described method of picking clonal organoids and Sanger sequencing, it would take Sanger sequencing results collected from hundreds of clones to make a convincing comparison of indel frequencies between PE3, HDR, and ABE. Alternatively, a high-throughput sequencing approach to quantify both correct editing and indels from pooled gDNA of all electroporated clones would provide sufficient evidence for the authors to compare editing efficiencies and indel frequency between PE3, HDR, and ABE in adult human stem cell-derived intestinal organoids.

A: We have included an extensive off-target analysis by first generating a clonal line, grown from a single cell from the intestinal line harboring the CFTR-R785 mutation. We have then electroporated this line with prime editing constructs and, based on FIS (swelling), we picked individual clones that have been genetically repaired. We then performed WGS on 5 repaired clones, where we do not see any unintended editing events outside of the targeted area. When we then compare mutational indel and single nucleotide signatures with unrepaired controls we do not see a significant increase in mutational load, pointing towards genome-wide safety of prime editing.*

Minor Points

1. Figures 1 and 2 should be combined, TP53 data is split across both figures and makes it difficult to follow the author's narrative.

A: Well taken. We have combined figure 1 and 2.

Reviewer #2 (Comments to the Authors (Required)):

1. A short summary of the paper, including description of the advance offered to the field. The current manuscript by Geurts and colleagues describes the application of a recently reported gene editing technology called prime editing to repair CFTR mutations. The new technology is discussed in comparison to the groups prior data using other CFRSPR editing technologies however do not directly show comparisons of the data in the manuscript. The manuscript reports on efficiency and off target undesired effects of prime editing using examples of insertion and correction of mutations. The manuscript provides a proof of principle study detailing prime editing as an editing tool with wider application than prior tested techniques. While the method may have potential clinical relevance, the technological advance is incremental and associated with inefficiencies and off target effects associated with the prime editing technology. These comparative observations, however, could be important to the field when considering use of the new technologies.

A: Thanks

2. For each main point of the paper, please indicate if the data are strongly supportive. If not, explicitly state the additional experiments essential to support the claims made and the timeframe that these would require.

The manuscript describes and compares the use of a new gene editing technology - called prime editing - for the functional repair of mutations in human intestinal cells. They apply the technology to 4 genes: 1) TP53, 2) PE3, 3) CFTR F508 and 4) CFTR R785.

The main take home messages are that

1) prime editing can be used to generate oncogenic mutations in intestinal organoids

a. The efficiencies reported in this manuscript seem to be dependent upon one experimental repeat in one biological specimen for each mutation and reports on the number of clones generated. This gives no indication of the efficiency - in the absence of editing and selection how many clones are usually observed? Is it possible that these mutations lead to complications with self-renewal and proliferation resulting in fewer clones?

A: We have now included experiments specifically to calculate editing efficiencies. For our experiments regarding modeling oncogenic mutations in TP53, we have co-transfected plasmids that contain genome editing components with PiggyBac plasmids that convey resistance towards Hygromycin. We then pick single clones that survive hygromycin selection and have thus been transfected. We performed sanger sequencing on 36 clones on in total 7 targets (R175H Liver versus intestine, R249S in liver VS intestine, Y220C base editing versus prime editing in intestine and C176F in intestine). Hereby we get a good understanding of the editing efficiencies on a diverse set of targets in multiple organs.

A: We do not expect that, in the modeling of oncogenic mutations in intestinal and hepatocyte organoids. Mutant organoids grow with the same kinetics as wt clones for each of the genes, and original clone sizes between wt, heterogous and homozygous mutant clones are essentially the same.

2) prime editing can be used for the functional repair of mutations causative for cystic fibrosis

a. Similarly, for the CFTR organoids the data seems to be from one patients' cells and an experimental repeat of 1 for each combination of reagents (Panel 3D). Given high patient to patient variability in these procedures it would be nice to see biological and experimental repeats to truly gauge the efficiency of the technique

A: We have now included an additional organoid line, derived from a patient harboring the CFTR-F508 del mutation and we have again counted repaired organoids. We do not see significant differences in the number of repaired organoids. Furthermore, for the modeling of oncogenic mutations we have included, instead of an additional intestinal organoid donor, hepatocyte organoids and we have compared editing efficiencies between the two different tissues.

3) Base editors are confirmed to be superior in terms of efficiency and specificity however are not able to target substantial regions of the genome. Prime editing is more versatile but is associated with lower efficiency and insertion of undesired edits on one or both alleles. It would be nice to have direct comparisons shown in the manuscript - perhaps the addition of a table would be very beneficial for the reader to compare the technologies.

A: This is a good suggestion, but we feel that this would be better suited for a review on the subject which would compose such a table based on multiple studies like the current one.

3. Lastly, indicate any additional issues you feel should be addressed (text changes, data presentation, statistics etc.)

The manuscript perhaps overstates the application of the new technology in the results sections based on the findings of the manuscript. It would be good to provide a more in-depth discussion of the technology's merits and issues in the discussion

A: We have substantially reworked the discussion of the paper with this suggestion in mind. We compare our results to other recent papers that have come out on prime editing in organoids such as Schene et al. 2020 Nature Communications and we discuss potential improvements to the technique that have recently come out such as Lin et al 2021 Nature Biotechnology and Liu et al 2021 Nature Communications.

June 25, 2021

RE: Life Science Alliance Manuscript #LSA-2020-00940-TR

Prof. Hans C. Clevers
Hubrecht Institute
Clevers group
Uppsalalaan 8
Utrecht, Utrecht 3584CT
Netherlands

Dear Dr. Clevers,

Thank you for submitting your revised manuscript entitled "Evaluating CRISPR-based Prime Editing for cancer modeling and CFTR repair in organoids". We would be happy to publish your paper in Life Science Alliance pending final revisions necessary to meet our formatting guidelines.

- please consult our manuscript preparation guidelines <https://www.life-science-alliance.org/manuscript-prep> and make sure your manuscript sections are in the correct order and labeled correctly
- please use the [10 author names, et al.] format in your references (i.e. limit the author names to the first 10)
- please revise the legend for Figure 1 (the actual figure doesn't have panels). Please be sure that legends follow figures and that there is no discrepancy between legend and introduced panels
- we encourage you to revise Figure 3 so that the panels appear alphabetically
- please add a panel C to Figure S4, as mentioned in the legend
- please revise callouts in the manuscript text (there are callouts for Figure 1D-F although these panels are not introduced in the actual figure nor the legend)
- please add callouts for Figures 4B, S4C to your main manuscript text
- please add a Data Availability Statement that includes relevant information for whole genome sequencing and mapping, as well as the link for the scripts used for mutation calling. These can still be mentioned where they currently appear in the Materials & Methods, but should also appear in this separate statement.

FIGURE CHECKS:

- please add scale bars to Figure 3E, 4C
- scale bars in Figure S2E are hardly readable

A. FINAL FILES:

B. MANUSCRIPT ORGANIZATION AND FORMATTING:

Thank you for this interesting contribution, we look forward to publishing your paper in Life Science

Alliance.

Sincerely,

Reviewer #1 (Comments to the Authors (Required)):

The authors have addressed the points raised in my initial review and I support publication of their revised manuscript.

July 16, 2021

RE: Life Science Alliance Manuscript #LSA-2020-00940-TRR

Prof. Hans C. Clevers
Hubrecht Institute
Clevers group
Uppsalalaan 8
Utrecht, Utrecht 3584CT
Netherlands

Dear Dr. Clevers,

Thank you for submitting your Research Article entitled "Evaluating CRISPR-based Prime Editing for cancer modeling and CFTR repair in organoids". It is a pleasure to let you know that your manuscript is now accepted for publication in Life Science Alliance. Congratulations on this interesting work.

DISTRIBUTION OF MATERIALS:

Again, congratulations on a very nice paper. I hope you found the review process to be constructive and are pleased with how the manuscript was handled editorially. We look forward to future exciting submissions from your lab.

Sincerely,
